# Non-invasive detection of human cardiomyocyte death using methylation patterns of circulating DNA

Hai Zemmour [1], David Planer[2], Judith Magenheim[1], Joshua Moss[1], Daniel Neiman[1], Dan Gilon[2], Amit Korach[3], Benjamin Glaser [4], Ruth Shemer[1], Giora Landesberg[5] & Yuval Dor[1]

Detection of cardiomyocyte death is crucial for the diagnosis and treatment of heart disease. Here we use comparative methylome analysis to identify genomic loci that are unmethylated specifically in cardiomyocytes, and develop these as biomarkers to quantify cardiomyocyte DNA in circulating cell-free DNA (cfDNA) derived from dying cells. Plasma of healthy individuals contains essentially no cardiomyocyte cfDNA, consistent with minimal cardiac turnover. Patients with acute ST-elevation myocardial infarction show a robust cardiac cfDNA signal that correlates with levels of troponin and creatine phosphokinase (CPK), including the expected elevation-decay dynamics following coronary angioplasty. Patients with sepsis have high cardiac cfDNA concentrations that strongly predict mortality, suggesting a major role of cardiomyocyte death in mortality from sepsis. A cfDNA biomarker for cardiomyocyte death may find utility in diagnosis and monitoring of cardiac pathologies and in the study of normal human cardiac physiology and development.

[1] Department of Developmental Biology and Cancer Research, The Institute for Medical Research Israel-Canada, The Hebrew University-Hadassah Medical School, Jerusalem 91120, Israel. [2] Department of Cardiology, Hadassah-Hebrew University Medical Center, Jerusalem 91120, Israel. [3] Department of Cardio-Thoracic Surgery, Hadassah-Hebrew University Medical Center, Jerusalem 91120, Israel. [4] Endocrinology and Metabolism Service, Hadassah-Hebrew University Medical Center, Jerusalem 91120, Israel. [5] Department of Anesthesiology and Critical Care Medicine, Hadassah-Hebrew University Medical Center, Jerusalem 91120, Israel. These authors contributed equally: Hai Zemmour, David Planer. Correspondence and requests for materials should be addressed to G.L. (email: giora.lan@mail.huji.ac.il) or to Y.D. (email: yuvald@ekmd.huji.ac.il)

Cardiac-specific troponins (cTn) are the gold standard for diagnosis of myocardial damage. However, these biomarkers do have limitations. Most importantly, it is difficult to determine if release of a cytoplasmic protein such as troponin to blood necessarily reflects cardiomyocyte death or may also occur also during reversible cellular injury[1]. Indeed, elevated troponins were measured after intensive exercise in healthy individuals[2,3], and some critically ill patients in surgical or medical intensive care units have significant troponin elevations. It is not known whether these elevations reflect myocardial cell death or reversible myocardial injury[4,5]. In addition, renal dysfunction slows down troponin clearance and significantly amplifies troponin levels[6], complicating the interpretation of troponin elevations in the context of renal failure, a common co-morbidity in patients with heart disease. Furthermore, since it is not clear how many protein molecules are released from each damaged cardiomyocyte, it is not possible to quantitate the number of injured cardiac cells in any clinical scenario. Thus, although cTN has proven clinical utility as a marker of myocardial damage, a biomarker capable of definitive quantification of cardiomyocyte death may complement current biomarkers and find utility in basic and clinical cardiology.

The blood contains nucleosome-size fragments of genomic DNA that are released from dying cells and circulate shortly before being cleared by the liver[7,8]. The analysis of cell-free, circulating DNA (cfDNA) is emerging as a powerful diagnostic modality for fetal chromosomal aberrations[9], cancer[10], and graft rejection[11,12], based on genetic differences between the host and the tissue of interest. These approaches are of limited use when cfDNA is derived from dying cells with a normal genome. The DNA of each cell type in the body carries unique methylation marks correlating with its gene-expression profile, representing a fundamental aspect of tissue identity[13]. We and others have recently described an approach to identify the tissue sources of cfDNA, based on tissue-specific methylation patterns in cfDNA[14–16], inferring cell death in tissues carrying a genome identical to the host.

Here we report the implementation of this technology for the study of human cardiomyocyte death. We established cardiomyocyte-specific cfDNA biomarkers, and applied them in two clinically relevant situations: acute ST-elevation myocardial infarction (STEMI) that is by definition associated with cardiomyocyte death, and sepsis or septic shock, where the existence and significance of irreversible cardiac damage has not been clearly established. Our preliminary data suggest that tracking cfDNA of cardiomyocyte origin provides clinically relevant diagnostic and prognostic information early in the course of sepsis.

## Results

**Identification of cardiomyocyte methylation markers**. To define genomic loci that are methylated in a cardiac-specific manner, we compared the methylomes of human heart chambers (right atrium, left and right ventricle) to the methylomes of 23 other human tissues, all publicly available[17]. Several differentially methylated loci were identified and a cluster of cytosines adjacent to the FAM101A locus (see online methods) was selected for further analysis (Fig. 1a, b). We PCR amplified a 90-bp fragment around this cluster after bisulfite conversion of unmethylated cytosines, and sequenced the PCR product to determine the methylation status of all six cytosines in the cluster. In purified cardiomyocyte DNA, 89% of the molecules were fully unmethylated, while in non-cardiac tissue <0.2% of molecules were unmethylated. In leukocytes (the main contributor to cfDNA), < 0.006% of molecules were unmethylated (Fig. 1c,

Supplementary Figure 1 and Supplementary Table 1). Thus the ratio of demethylated molecules in heart:blood DNA was 89:0.006 giving a theoretical signal to noise ratio of 15,000. In DNA of skeletal muscle and colon, 0.1–0.2% of FAM101A molecules were unmethylated, providing a heart:skeletal muscle/colon signal ratio above 445 (Supplementary Figure 1 and Supplementary Table 1). While these findings indicate an excellent cardiac specificity of FAM101A methylation in the tissue panel examined, we acknowledge that DNA from additional human tissues and cell types must be tested to further rule out potential cross-reactivity.

To determine the linearity and sensitivity of the assay, we performed spike-in experiments. Cardiac and leukocyte DNA were mixed in different proportions, and the fraction of cardiac DNA in the mixture was assessed using PCR amplification and massively parallel sequencing. The assay correctly determined the fraction of cardiac DNA, even when it was only 0.1% of the DNA in the mixture (Fig. 1d, e), and cardiac DNA could be detected robustly when ten or more cardiac genomes were present (Supplementary Figure 1).

Following bisulfite treatment, DNA becomes single stranded. Therefore, both strands can be measured in a solution, as if they were independent biomarkers. To test this idea, we designed primers against the antisense strand of FAM101A post-bisulfite conversion. As expected, the sense and antisense templates showed similar sensitivity and specificity (Fig. 1b–e and Supplementary Figure 1). We reasoned that by testing both strands in a given sample, both sensitivity and specificity of the assay would increase. Therefore, further analysis of clinical samples was performed with multiplex PCR using both sense and antisense specific primer sets.

**Plasma levels of cardiomyocyte DNA in healthy individuals**. We used the sense and antisense FAM101A markers to assess the concentration of cardiac cfDNA in the plasma of donors. We extracted cfDNA from plasma, treated with bisulfite and performed PCR followed by sequencing, typically using material from 0.5 ml of plasma. The fraction of PCR products carrying the cardiac-specific methylation pattern was multiplied by the total concentration of cfDNA, to obtain an estimation of cardiac cfDNA content in plasma.

Plasma from 83 healthy adult volunteers was tested and 0–1 copies of cardiac cfDNA/ml plasma were detected in 60 (Fig. 2a). This low level of a signal likely reflects the extremely low rate of cardiomyocyte death in healthy adults[18]. In 23 volunteers, 2–14 copies of cardiac cfDNA/ml plasma were found. This weak signal may reflect minor cross reactivity with DNA from another tissue, an underlying cardiac abnormality in these volunteers, or occasional, minor PCR contamination. The mean plus three standard deviations of the control group was ten copies/ml. We therefore defined this level as the cutoff for a positive signal.

**Cardiomyocyte cfDNA in acute myocardial infarction**. As a positive control where high levels of cardiac cfDNA are expected, we used plasma from patients with acute STEMI. We obtained samples from STEMI patients upon hospital admission before primary percutaneous coronary intervention (PCI), shortly after they underwent PCI, and again up to 56 h after admission. We measured the levels of cardiac cfDNA as well as the plasma levels of high-sensitivity troponin-T (hs-cTn) and CPK.

First, we asked whether our cfDNA biomarker was at all able to identify evidence of cardiomyocyte death. We found that STEMI patients had dramatically higher levels of cardiac cfDNA compared with healthy controls (Fig. 2a). To assess assay performance in discriminating healthy from all STEMI plasma samples we plotted Receiver Operator Characteristic curve. The

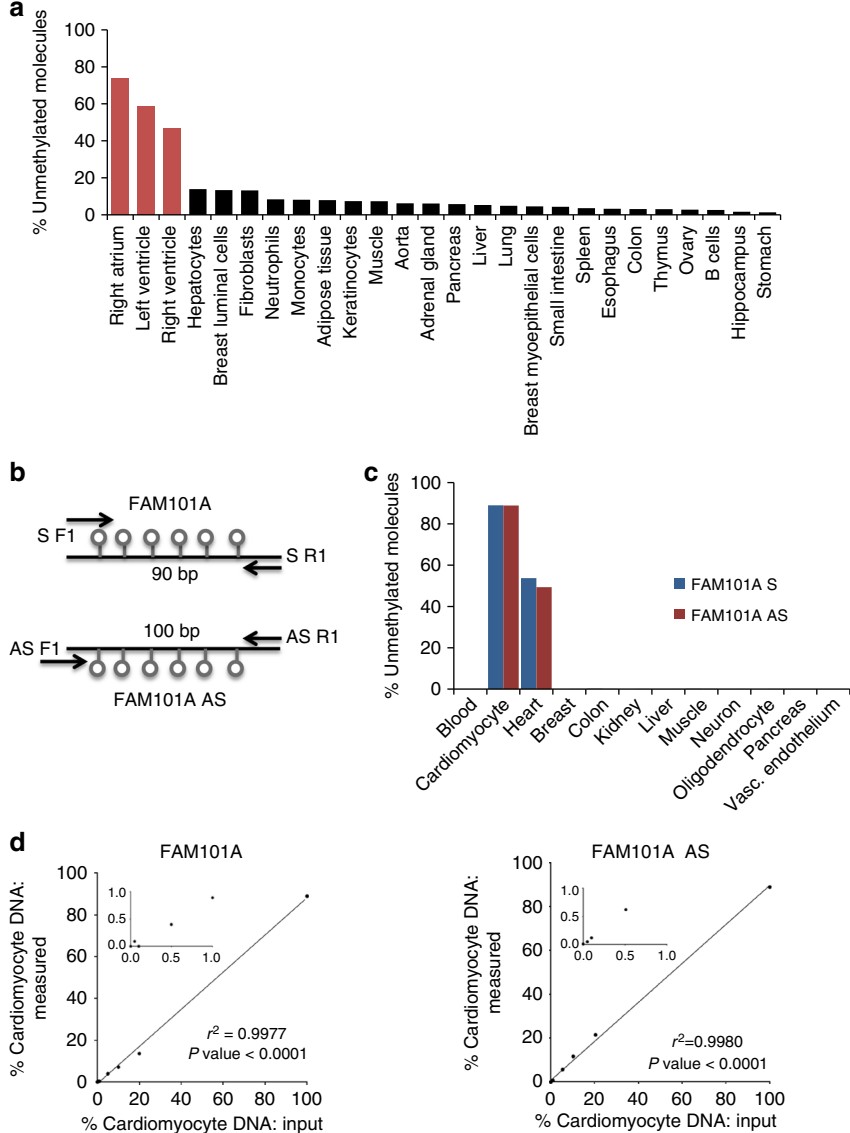

**Fig. 1** Identification of cardiomyocyte-specific DNA methylation markers. **a** Unmethylation levels of *FAM101A* locus in 27 human tissues, including left ventricle, right ventricle and right atrium (red). Data representing methylation status of all CpG sites in the locus was extracted from the Roadmap Epigenomics Consortium browser. **b** Structure of the *FAM101A* locus, used as two independent markers: FAM101A and FAM101A AS. Lollipops represent CpG sites; arrows mark positions of PCR primers; S sense marker, AS antisense marker. **c** Demethylated FAM101A and FAM101A AS in DNA from multiple tissues and from isolated cardiomyocytes (purchased from ScienCell Research Laboratories, San Diego, CA). See online methods for an explanation why targeted PCR yields a lower background in non-cardiac tissues compared with the Roadmap browser in panel **a**. **d** Spike in experiments for FAM101A (left) and FAM101A AS (right). Human cardiomyocyte DNA was mixed with human leukocyte DNA in the indicated proportions (0–100%), and the percentage of fully unmethylated *FAM101A* molecules (in which all CpG sites were converted by bisulfite) was determined

area under the curve (AUC) was 0.94, indicating high sensitivity and specificity (Fig. 2b). Interestingly, cardiac cfDNA levels were increased in STEMI patients with normal (<200) CPK levels (Fig. 2c) suggesting greater sensitivity. In contrast, cardiac cfDNA was elevated in only one of six STEMI patients with normal (<0.03) hs cTn levels (Fig. 2d).

A comparison of troponin levels to cardiac cfDNA in 57 samples from STEMI patients yielded Spearman correlation value of 0.79 and *p*-value < 0.0001 (Fig. 2e). When plotting cardiac cfDNA vs troponin and marking on each axis the threshold of a positive signal, 79% of the STEMI samples were positive for both troponin and cardiac cfDNA, 7% were negative for both, 11% were positive only for troponin, and 4% were weakly positive only for cardiac cfDNA (Fig. 2f). The discordant

samples may reflect differences in the rates of release and clearance of troponin and cfDNA, or alternatively, cases where troponin leak was not accompanied by major cardiomyocyte death.

Importantly, unlike cardiac-specific cfDNA, the total cfDNA concentration in STEMI did not correlate with troponin or CPK, nor with the percentage of cardiac cfDNA (Supplementary Figure 2). This reflects the fact that total cfDNA integrates all recent cell death events, including contributions from other tissues, thereby masking the cardiac-specific signal, and emphasizes the importance of calculating the specific contribution of the heart to cfDNA when assessing cardiac damage. The sense and antisense markers correlated well in the STEMI plasma samples (Supplementary Figure 2).

Having determined that cardiac cfDNA levels can indeed detect cardiomyocyte damage, we went on to examine the kinetics of cardiac cfDNA at serial time points. As can be seen in Fig. 3a, cardiac cfDNA signal was sufficient to distinguish people with MI prior to intervention (0–2 h after onset of chest pain) from healthy individuals (AUC = 0.8117, p-value < 0.001, Mann–Whitney test, Fig. 3b). Coronary revascularization in STEMI patients is known to cause an abrupt increase of cardiac markers, an indication of successful reperfusion. Indeed, cardiac cfDNA increased dramatically in most patients after PCI (Fig. 3a and Supplementary Figure 3), further supporting authenticity of the signal. A more detailed time course on representative patients revealed that cardiac cfDNA levels rose quickly after PCI and returned to baseline after 1–2 days, while troponin levels remained high (Fig. 3c and Supplementary Figure 3).

We conclude that measurements of cardiac cfDNA capture cardiomyocyte cell death associated with myocardial infarction, and that the cardiac cfDNA assay can identify myocardial cell death early after ischemia ensues. Additional more detailed clinical studies are required to determine how cardiac cfDNA compares to troponin in predicting infarct size and long-term cardiac function.

**Cardiomyocyte cfDNA in patients with sepsis.** Patients with sepsis or septic shock often have elevated levels of troponin[19], although the cause of cardiac injury and whether it represents true myocardial cell death or transient stress and troponin leak absent of cell death is not known[20–22]. In addition, since troponin is cleared via the kidneys, renal dysfunction, common in septic patients, may contribute to elevated troponin[6]. Because cfDNA is a more definitive marker of cell death and is cleared by the liver[23], we reasoned that measurements of cardiac cfDNA could be informative in this setting.

We determined the levels of cardiac cfDNA in a cohort of 100 patients with sepsis or septic shock, for which 201 plasma samples were available[22]. Cardiac cfDNA was assessed blindly, and values were correlated with cardiac troponin and clinical parameters.

Septic patients had high levels of total cfDNA, reflective of broad tissue damage (Supplementary Figure 4), as previously reported[24]. Strikingly, many patients also had high levels of cardiac cfDNA, similar in magnitude to the acute setting of STEMI (Fig. 4a). These findings suggest that significant cardiomyocyte death is common in septic patients. The sense and antisense markers of FAM101A correlated well, supporting specificity of the signal (Supplementary Figure 4). Cardiac cfDNA

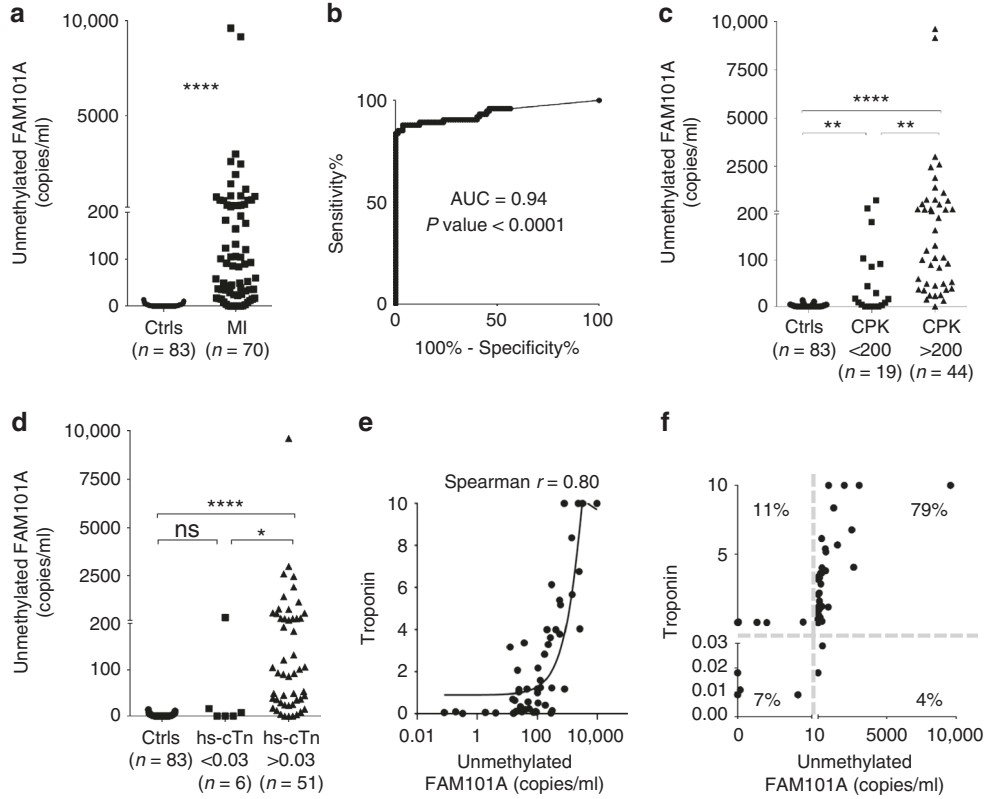

**Fig. 2** Cardiac cfDNA in healthy subjects and in patients with myocardial infarction. **a** Cardiac cfDNA (copies of fully unmethylated *FAM101A* /ml plasma) in samples from healthy controls (Ctrls, n = 83) and patients during STEMI (MI, n = 74 samples from 31 patients). Mann–Whitney test for controls vs. patients, p-value < 0.0001. **b** Receiver operating characteristic (ROC) curve for the diagnosis of STEMI by demethylated *FAM101A* in plasma of healthy controls and patients with MI. Area under the curve (AUC) 0.94 (95% CI = 0.9044 to 0.983), p-value < 0.0001. **c** Comparison of unmethylated *FAM101A* levels (copies/ml) in healthy controls, STEMI samples with normal Creatine Kinase (CPK < 200) and STEMI samples with high CPK (CK > 200). Kruskal-Wallis test p-value < 0.0001. Dunn's multiple comparisons test: Controls (Ctrls) vs. STEMI with normal CPK, p-value < 0.001; Ctrls vs. STEMI with high CPK, p-value < 0.0007; STEMI with normal CPK vs. STEMI with high CPK, p-value = 0.0068. **d** Comparison of unmethylated *FAM101A* levels in healthy controls, samples from STEMI patients with normal levels of high-sensitive cardiac troponin T (hs-cTn) (<0.03), and samples from STEMI patients with high levels of hs-cTn (>0.03). Kruskal–Wallis test p-value < 0.0001. Dunn's multiple comparisons test: Ctrls vs. STEMI with normal hs-cTn, p-value = 0.6863; Ctrls vs. STEMI with high hs-cTn, p-value < 0.0001; STEMI with normal hs-cTn vs. STEMI with high hs-cTn (>0.03), p-value = 0.0307. **e** Spearman correlation between cardiac cfDNA and troponin levels in 57 STEMI patients. Curved line, non linear (quadratic) fit. **f** XY Scatter plot for cardiac cfDNA levels vs. cardiac troponin. Quadrants indicate negative and positive hs-cTn, and negative and positive cardiac cfDNA. Numbers indicate the percentage of samples in each quadrant

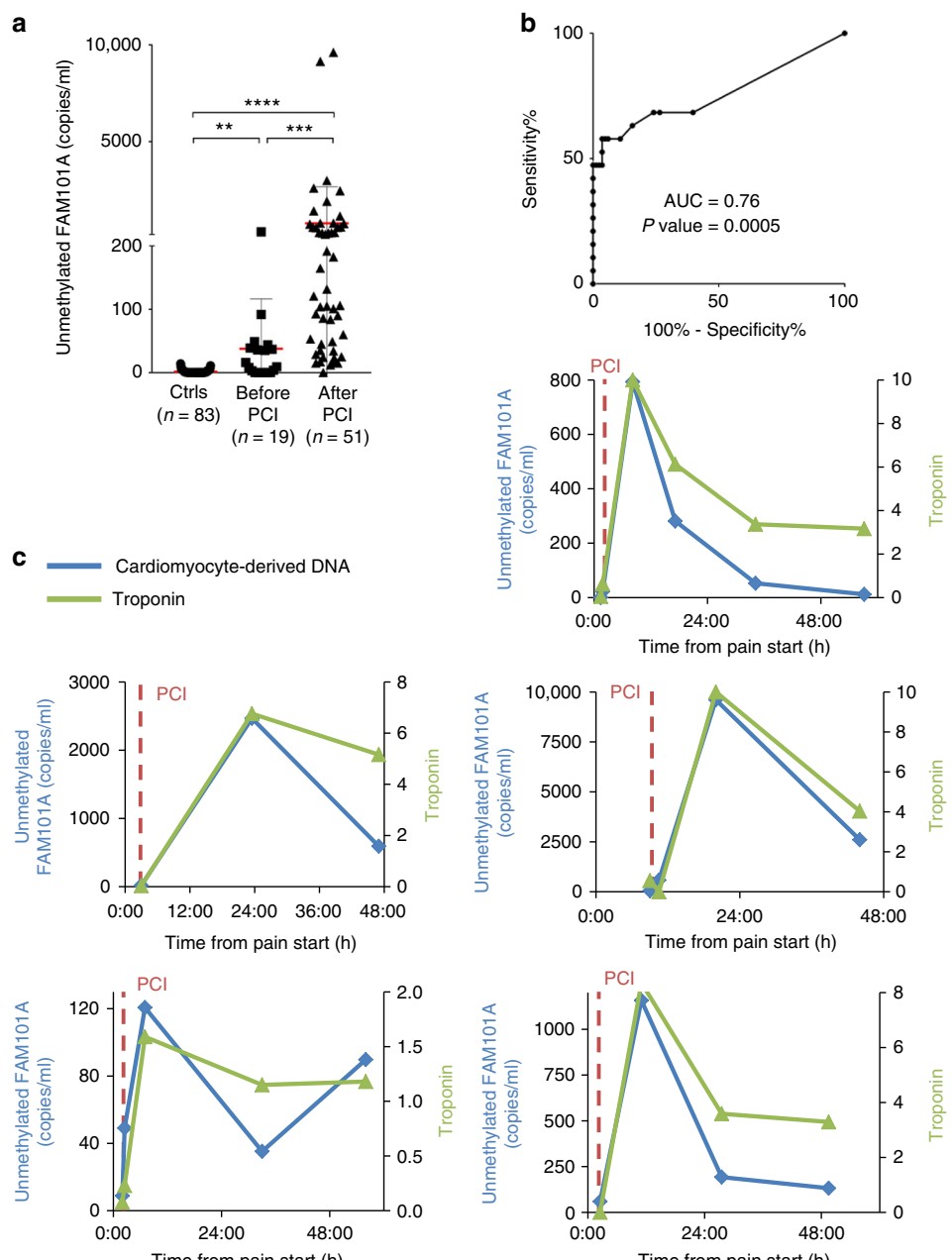

**Fig. 3** Cardiac cfDNA dynamics during STEMI and after angioplasty. **a** Comparison of unmethylated *FAM101A* levels in healthy controls, and STEMI samples before and after PCI. Kruskal–Wallis test *p*-value < 0.0001. Dunn's multiple comparisons test: Ctrls vs. pre-PCI, *p*-value = 0.0082; Ctrls vs. post-PCI (>0.03), *p*-value < 0.0001; per- vs post-PCI, *p*-value = 0. 0005. Horizontal red and gray lines represent the average value and standard deviation of cardiac cfDNA among the samples from each group. **b** ROC curve for the diagnosis of STEMI by cardiac cfDNA in healthy individuals versus STEMI patients prior to intervention. Area under the curve (AUC) 0.76, *p*-value = 0.0005. **c** Time course of cardiac cfDNA and troponin levels in representative individual patients. Vertical dashed lines indicate PCI time

and troponin levels did not correlate in sepsis, unlike the situation in STEMI (Fig. 4b). The difference may arise from the distinct pathophysiology of myocardial damage in sepsis as opposed to myocardial infarction or from differences in overall clinical status. While STEMI is an abrupt coronary event, myocardial damage in sepsis is likely a prolonged event occurring over a period of days. Although blood samples were obtained as close to admission to the intensive care unit as possible, this time point is random relative to the peak of myocardial cell damage. In addition, differential clearance rates of troponin and cfDNA potentially have an impact on biomarker measurements in this setting. Cardiac cfDNA (and total cfDNA) elevation in septic patients did

not correlate with liver damage markers AST and ALT nor with creatinine levels (Supplementary Figure 5), supporting the idea that cardiac cfDNA mainly reflects cardiac damage and is not significantly altered by clearance rate of cfDNA due to liver or kidney dysfunction.

Cardiac cfDNA elevation strongly predicted short term mortality in sepsis (Fig. 4c). Septic patients with elevated cardiac cfDNA were four times more likely to die within 90 days of hospitalization than patients with no cardiac cfDNA. Cardiac cfDNA predicted mortality stronger than troponin, stronger than total cfDNA and independent of age (Cox's multivariate survival analysis: Supplementary Table 2). These findings indicate that

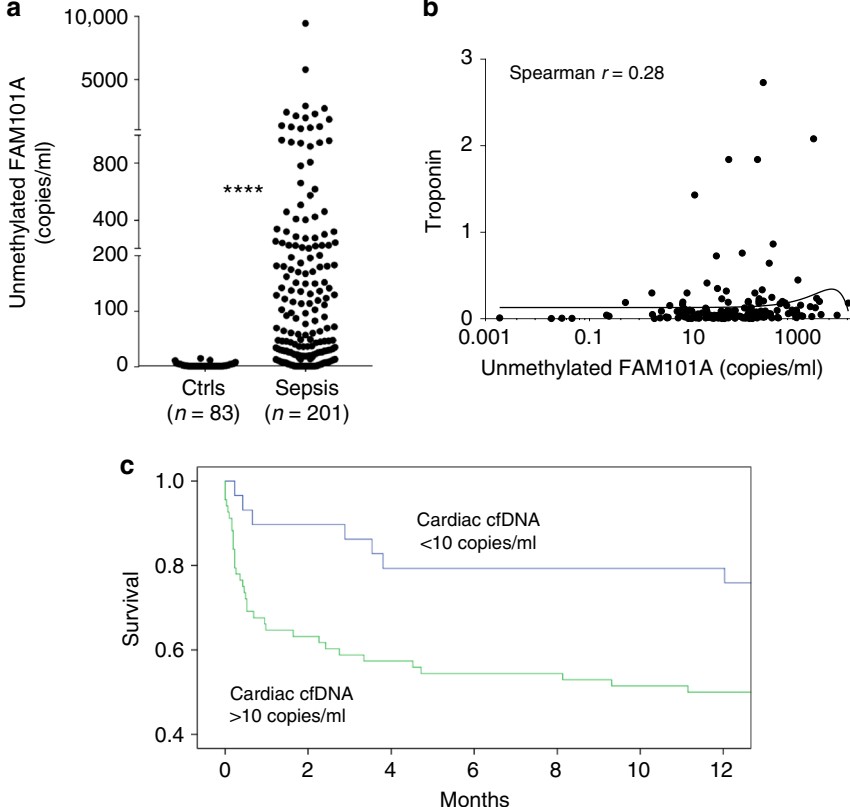

**Fig. 4** Cardiac cfDNA in sepsis. **a** Levels of cardiac cfDNA in healthy controls ($n = 83$) and patients with sepsis ($n = 201$). Mann–Whitney test for controls vs. patients, $p$-value < 0.0001. **b** Lack of correlation between cardiac cfDNA and troponin. Curved line represents non-linear (quadratic) fit. Spearman $r$ coefficient of correlation was calculated. **c** Kaplan–Meier plot showing correlation of cardiac cfDNA to patient survival

myocardial cell death is an important determinant of mortality following sepsis, and that elevated cardiac cfDNA is a strong prognostic biomarker in sepsis.

**A digital droplet PCR (ddPCR) procedure for measuring cardiac cfDNA.** The assay described above, relying on next generation sequencing, takes >24 h from drawing blood to results, which hinders utility in emergency medicine. We therefore sought to measure cardiac cfDNA using a more rapid PCR format. We established a procedure using ddPCR to accurately count the number of molecules carrying the cardiac methylation signature at the FAM101A locus. Standard ddPCR methods are capable of interrogating only one or two CpGs on a molecule. Since the specificity and sensitivity of our marker requires interrogation of multiple CpGs on a single molecule[14], we designed the assay to simultaneously interrogate five CpGs in the locus using two fluorescent probes, each capturing distinct two or three unmethylated cytosines (Fig. 5a and Material and Methods), leveraging the increased specificity attributed to regional methylation status[14].

ddPCR analysis of cardiomyocyte and leukocyte DNA revealed that each probe alone was able to discriminate between DNA from the two sources, with a signal to noise ratio of 50:1 to 58:1. However, when we scored only droplets positive for both probes, the cardiomyocyte:leukocyte signal ratio increased to 258, affording a five-fold increase in specificity (Fig. 5b). ddPCR on cardiac DNA spiked into leukocyte DNA gave a signal that increased linearly with the amount of cardiac DNA; scoring only dual-labeled probes gave a lower baseline signal than scoring individual probes, better reflecting cardiomyocyte contribution to the mixture (Figure 5c).

Finally, we tested the ddPCR strategy on plasma samples. ddPCR revealed a clear signal in the plasma of STEMI patients and was able to distinguish well between controls and patients. A lower baseline signal was observed in healthy individuals when scoring only dual-labeled probes, indicating increased specificity (Fig. 5d). We conclude that the ddPCR assay for cardiac cfDNA provides a rapid and simple alternative to sequencing-based assays.

## Discussion

We propose that cardiac cfDNA measurements can become a new, specific marker for myocardial damage, potentially complementing currently available markers. First, the presence of cardiac cfDNA may help distinguish cardiomyocyte death from reversible damage. This is of interest in various physiological and pathological scenarios, for example myocardial strain post intensive exercise, secondary myocardial injury in acute illness, or hibernating ischemic myocardium. In these scenarios troponin elevation may occur result from reversible myocardial injury and leak from the cytoplasmic pool, without actual cell necrosis[21,25]. If cfDNA exclusively reflects cell death, as is commonly believed, cardiac cfDNA is expected to remain normal in conditions of reversible heart damage. We acknowledge however that a tight link between cfDNA and cell death has to be formally demonstrated, since cells could in theory release DNA fragments without dying[26–29]. Additional experiments are needed to test the idea that cardiac cfDNA is an exclusive marker of cardiomyocyte death. If this turns out to be the case, then the known content of DNA per cell may allow a quantitative inference of the rate of cell death. Second, cfDNA is mostly cleared by the liver[23], making cardiac (as well as total) cfDNA measurements more resistant to

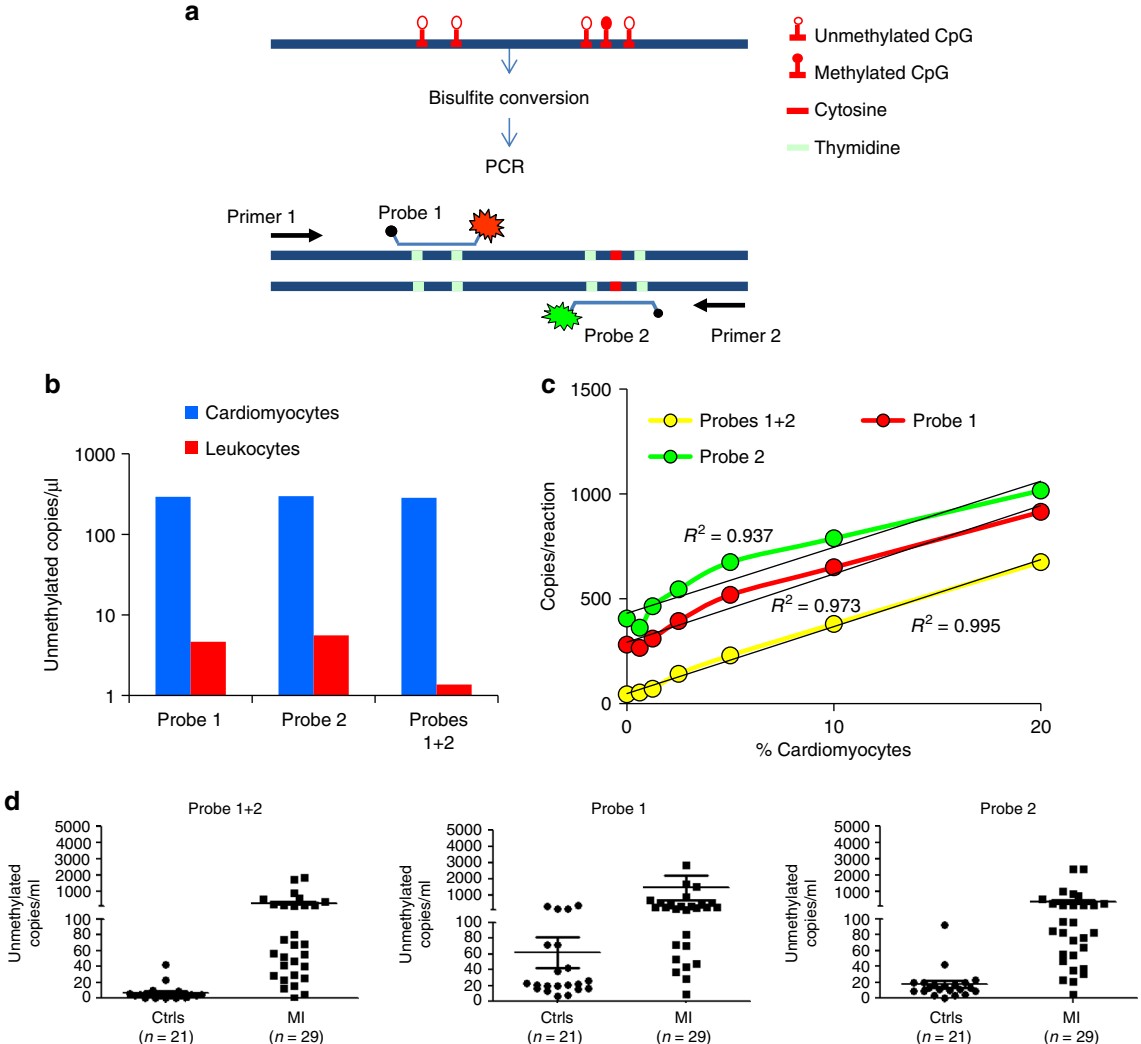

**Fig. 5** Detection of cardiac cfDNA using digital droplet PCR. **a** Schematic of approach for ddPCR-based detection of methylation status of multiple adjacent cytosines. A signal from two probes in the same droplet reflects lack of methylation in five adjacent cytosines in the same original DNA strand. **b** Signal from cardiomyocyte and leukocyte DNA based on individual or dual probes. Scoring only dual probe signals drastically reduces noise from leukocyte DNA. **c** Spike-in experiment assessing sensitivity and linearity of signal from cardiomyocyte DNA diluted in leukocyte DNA. The use of dual probe enhances linearity and reduces baseline signal. *x* axis shows both the % of cardiac DNA diluted into blood DNA, and the absolute number of cardiomyocyte genomes present in each sample. **d** Measurement of cardiac cfDNA in plasma of healthy adult and patients with myocardial infarction. The use of dual probes reduces the baseline signal in healthy plasma. Horizontal lines represent average and standard deviation of cfDNA values among the samples in each group

kidney dysfunction[30,31], a common confounder in troponin assays. Empirically, our observations in septic patients suggest that the levels of cardiac cfDNA were not significantly affected by either renal or hepatic damage. Third, cfDNA is thought to be cleared within 15–60 min, faster than troponin or CPK[32]. This suggests that cardiac cfDNA reflects the rate of cardiomyocyte death close to the time of sampling, complementing the longer-term integration of cardiac damage provided by troponin and CPK. Experiments to assess the clearance rate of cfDNA in specific conditions such as sepsis will be needed to test this idea. cfDNA measurements might be particularly helpful in dynamic situations, for example sequential or on-going myocardial insults.

We foresee major improvements in measurements of cardiac cfDNA in terms of time to results, specificity and sensitivity, as understanding of methylation biology expands and DNA analysis technology improves.

Some applications of diagnostics in cardiology are time-sensitive. Our ddPCR version of the cardiac cfDNA assay yields a turnaround time of 6.5–7 h from drawing blood to obtaining results. Further optimization can reduce this time considerably, and upcoming technologies such as nanopore sequencing[33] may allow for near real-time measurement of cardiac cfDNA.

Regarding tissue specificity, FAM101A methylation differentiates cardiomyocytes from all tissue tested including skeletal myocytes, showing specificity superior to the general muscle markers CPK and myoglobin, and similar to cardiac troponin. More cell types need to be tested to confirm that no other tissue damage can result in a positive cardiac cfDNA signal. Future studies will investigate methylation signatures of cardiac sub-domains and cellular sub-types, opening a window into non-invasive detection of damage in specific areas of the heart.

Regarding sensitivity, the current cfDNA assay appears slightly inferior to troponin. Multiplexing independent cardiac methylation markers may dramatically increase sensitivity (that is, the chance that a heart-derived DNA fragment is successfully captured). We also note that unlike protein markers, cfDNA marker sensitivity increases when a larger volume of blood is drawn.

In the context of sepsis, our findings provide insights beyond assay validation. First, our results strongly support the notion that cardiomyocyte death occurs in sepsis. Second, the strong correlation between cardiac cfDNA and mortality not only provides a useful prognostic biomarker, but also underscores the importance of cardiac health in facilitating survival from sepsis. These findings may have implications for the management of septic patients.

The cardiac cfDNA assay may find broad utility in the study of human heart biology. For example, it will be possible to determine the baseline levels of cardiac cell death from birth to old age[18] to better model human heart development. Another attractive application is in sports medicine, where our assay may provide insights into the nature of elevated troponin in the plasma of athletes, a puzzling observation with unknown long-term consequences[25]. We also note that cfDNA is already being used for early detection of cardiac transplant rejection[12], using donor-specific SNPs in recipient circulation. Our cardiac cfDNA assay does not require tailoring to specific donor-recipient genetics, and measures specifically cardiomyocyte death (avoiding, for example, a signal from death of donor fibroblasts). These features of methylation-based cfDNA detection may be used broadly in the transplant rejection setting.

## Methods

**Clinical samples.** The Hadassah–Hebrew University Medical Center Institutional Review Board approved all clinical studies and informed consent was obtained from all human participants accordingly.

For the myocardial infarction study, thirty-one patients with the clinical diagnosis of STEMI, who presented to the Hadassah Hospital emergency service due to acute chest pain, were recruited to participate in the study. Patients were enrolled if they presented within 12 h after the onset of symptoms and had ST-segment elevation of >1 mm in two or more contiguous leads. All patients were immediately taken to the catheterization lab for coronary angiography and primary PCI as indicated. Final diagnosis was determined after coronary angiography was performed. Patients were diagnosed with STEMI if they had coronary obstruction / thrombus in the culprit coronary artery.

The time from the onset of chest pain to first sampling (prior to PCI) was: within 1 h from chest pain, $n = 3$; 1–4 h, $n = 17$; 4–8 h, $n = 5$; 8–13 h, $n = 6$. Blood samples were obtained in EDTA containing tubes before PCI, immediately after PCI, and at 6-h intervals, up to 60 h from hospital admission. Emergency coronary angiography followed by PCI, when appropriate, was performed according to standard clinical practice. Plasma was separated as described below within 2 h of ascertainment, transferred to clean tubes and frozen at −80 °C until assay.

For the sepsis study, patients studied were a randomly selected subset of our previously published study that was designed to investigate the relationships between myocardial dysfunction on echocardiography and myocardial injury as detected by serum troponin elevations in patients with sepsis and septic shock[22]. In brief, these were critically ill, mechanically ventilated patients treated in the intensive care unit of Hadassah Hospital for sepsis or septic shock. All clinical, demographic, laboratory and survival data were prospectively collected. Blood samples were obtained repeatedly shortly after their admission to the intensive care unit, centrifuged and stored at −80 °C until analysis.

A total of 83 adult volunteers participated in the study as unpaid healthy controls. The age distribution of these healthy donors was 18–66 years, average 35.3. The sex distribution was 44 females, 37 males; for two donors information was not available. All donors serving as controls denied having any acute or chronic illnesses or receiving any medications at the time of blood donation.

The volume of blood samples was as follows. When obtaining fresh samples, 20 ml of blood were drawn, from which 8 ml of plasma were obtained; 4 ml plasma were used for extraction of cfDNA. When working with archived material (most of the sepsis samples), we used no less than 400 µl plasma. Specifically, for the MI samples we used 1000–4000 µl, average 3721; Sepsis samples: 400–3500, average 1804; and control samples: 800–4000, average 3043.

**Protein biomarkers.** High-sensitivity troponin-T was measured using Roche Diagnostics, Elecsys Assays, the normal values of which is <0.03 ng/mL. CPK was measured by Roche Diagnostics, Cobas C Systems, with normal values <200 U/L.

**Identification of cardiac methylation markers.** Potential cardiomyocyte specific biomarkers were selected by comparing methylation levels of CpGs throughout the genome using publically available DNA methylation data sets generated by whole-genome bisulfite sequencing (Roadmap Epigenomics Consortium, and the German Epigenom Programm –DEEP). The genome was scanned for <100 bp regions

containing five or more CpGs, which were methylated (>80%) in all non-cardiac tissues, and unmethylated in heart tissues (right atrium, left and right ventricle). The fragment of FAM101A, located in chromosome 12 with coordinates 124692462–124692551, was selected for further investigation. We first calculated the percentage of fully unmethylated FAM101A molecules (six CpGs on this 90 bp fragment) in non-cardiac tissues using the Roadmap browser (Fig. 1a). We expected that this approach would over-estimate the percentage of fully unmethylated molecules since in the Roadmap data whole genome bisulfite sequencing was used, and average levels of demethylation were calculated from all molecules sequenced. Consequently, many molecules in non-cardiac tissues were scored as unmethylated even though, using this method, it was impossible to know how many of these in fact represented individual molecules in which all six CpGs were unmethylated. In our PCR-based targeted approach (see below), we scored the methylation status of all CpGs on individual molecules, counting only those molecules on which all CpGs were unmethylated. As shown in Fig. 1c, Supplementary Figure 1 and Supplementary Table 1, fully unmethylated molecules were rarely found outside the heart.

**cfDNA analysis.** Blood samples were collected in EDTA tubes, and centrifuged at 1500$g$ for 10 min at 4 °C within 2 h of collection. Plasma was removed and re-centrifuged at 3000$g$ for 10 min at 4 °C to remove any remaining cells. Plasma was then stored at −80 °C until assay. Cell-free DNA was extracted using the QIA-symphony SP instrument and its dedicated QIAsymphony Circulating DNA Kit (Qiagen) according to the manufacturer's instructions. DNA concentration was measured using the Qubit® dsDNA HS Assay Kit (Thermo Fisher Scientific). cfDNA was treated with bisulfite using EZ DNA Methylation-Gold™ (Zymo Research), and PCR amplified with primers specific for bisulfite-treated DNA but independent of methylation status at the monitored CpG sites. Treatment with bisulfite led to degradation of 60–90% of the DNA (on average, 75% degradation), consistent with previous reports[34]. Note that while DNA degradation does reduce assay sensitivity (since fewer DNA molecules are available for PCR amplification), it does not significantly harm assay specificity since methylated and unmethylated molecules are equally affected. Primers were bar-coded using TruSeq Index Adapters (Illumina), allowing the mixing of samples from different individuals when sequencing PCR products using MiSeq or NextSeq sequencers (Illumina). Sequenced reads were separated by barcode, aligned to the target sequence, and analyzed using custom scripts written and implemented in R. Reads were quality filtered based on Illumina quality scores, and identified by having at least 80% similarity to target sequences and containing all the expected CpGs in the sequence. CpGs were considered methylated if *CG* was read and were considered unmethylated if *TG* was read. The fraction of unmethylated molecules in a sample was multiplied by the total concentration of cfDNA in the sample, to assess the number of cardiac genome equivalents per ml of plasma. The concentration of cfDNA was measured prior to bisulfite conversion, rendering the assay robust to potential inter-sample fluctuations in the extent of bisulfite-induced DNA degradation.

**Digital droplet PCR.** We established a procedure for ddPCR, in which bisulfite-treated cfDNA is interrogated using two methylation-sensitive TaqMan™ probes. The limited length of probes (up to 30 bp) dictated that they could cover only two or three informative CpG sites in the FAM101A locus, predicting a relatively high frequency of noise (positive droplets) in DNA from non-cardiac tissue (Fig. 5b). In the sequencing-based assay we addressed this problem by documenting the methylation status of 6 adjacent cytosines (Fig. 1), which greatly increased specificity. To implement this concept in the ddPCR platform we designed two TaqMan probes, each recognizing lack of methylation in a different cluster of cytosines (one containing two CpG sites and one containing three CpG sites) within the same amplified 100 bp fragment from the FAM101A locus (Fig 5a). Each probe was labeled with a different fluorophore, such that we could identify droplets in which both probes found a target. Such droplets would be interpreted as containing a FAM101A cfDNA fragment in which all five targeted cytosines were unmethylated. This resulted in a ddPCR assay with the improved specificity afforded by interrogating multiple cytosines on the same DNA molecule (Fig. 5).

The following primers were used for the analysis of five cytosines located adjacent to the FAM101A locus: PCR primers 5′-TATGGTTTGGTAATTTATT TAGAG-3′ (forward) and 5′-AAATACAAATCCCACAAATAAA-3′ (reverse) in combination with probes that detected lack of methylation on three and two cytosines, respectively: 5′-AATGTATGGTGAAATGTAGTGTTGGG-3′ (FAM-forward probe) and 5′-AAAAATACTCAACTTCCATCTACAATT -3′ (HEX-reverse probe). Assay design is shown in Fig. 5a. Each 20-µL volume reaction mix consisted of ddPCR™ Supermix for Probes (No dUTP) (Bio-Rad), 900 nM primer, 250 nM probe, and 2 µL of sample. The mixture and droplet generation oil were loaded onto a droplet generator (Bio-Rad). Droplets were transferred to a 96-well PCR plate and sealed. The PCR was run on a thermal cycler as follows: 10 min of activation at 95 °C, 47 cycles of a two-step amplification protocol (30 s at 94 °C denaturation and 60 s at 53.7 °C), and a 10-min inactivation step at 98 °C. The PCR plate was transferred to a QX100 Droplet Reader (Bio-Rad), and products were analyzed with QuantaSoft (Bio-Rad) analysis software. Discrimination between droplets that contained the target (positives) and those that did not (negatives) was

achieved by applying a fluorescence amplitude threshold based on the amplitude of reads from the negative template control.

**Code availability**. Custom script for sequence analysis is available from the authors upon reasonable request.

**Data availability**. All relevant data including primer sequences, detailed PCR conditions and additional protocols are available from the authors upon reasonable request.

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

## Acknowledgements
The authors thank Shoshi Shpitzen, Ofri Abraham, David Rosenblatt, Sheina Piyanzin and Ibrahim Knani for excellent technical help, Lilach Gavish for coordination of sample collection, Howard Cedar, Shmuel Ben-Sasson and Aharon Razin for stimulating discussions, and the patients and healthy donors who participated in the study. Supported by grants from the Kahn Foundation and the NIDDK-supported Human Islet Research Network (HIRN, RRID:SCR_014393; UC4 DK104216-01, to YD). Supported in part by a grant from USAID's American Schools and Hospitals Abroad Program for the upgrading of the Hebrew University Medical School Sequencing Facility.

## Author contributions
Conceptualization—D.P., D.G., B.G., R.S., G.L., Y.D.; Investigation—H.Z., D.P., Ju.M., Jo. M., D.N.; Resources—D.P., A.K., G.L.; Writing—H.Z., D.P., Ju.M., Jo.M., B.G., R.S., G.L., Y.D.

## Additional information

**Competing interests:** Ju.M., R.S., B.G. and Y.D. are inventors on a patent entitled "A Novel approach for improving the detection of tissue-specific DNA: Sense and Antisense strands of bisulfite treated DNA" (patent number 62/531,990), and a patent entitled "A dual-probe digital droplet PCR strategy for specific detection of tissue-specific circulating DNA molecules" (patent number 62/531,983); H.Z., Ju.M., Jo.M., D.N., R.S., B.G. and Y. D. are inventors on a patent entitled "New set of probes for determining tissue of origin by methylation pattern" (patent number 62/531,988). The remaining authors declare no competing interests.

