## [Peer Review File · Nature Communications]

Editorial Note: Parts of this Peer Review File have been redacted as indicated, as we could not obtain permission to publish the reports of reviewer #4. During the second round of review, reviewer #4 expressed concerns that were addressed in the final round of revision

Reviewer #1:

Remarks to the Author:

Zemmour et al report their investigation of using cardiomyocyte-associated methylation signatures to develop plasma DNA markers for detecting human cardiomyocyte cell death. This work follows on publications by this team and other groups that circulating methylation markers may serve as laboratory markers for different clinical conditions.

I have the following specific comments.

1. Treatment with bisulfite has generally been known to be destructive to DNA. How significant is such degradation on the robustness of detecting circulating cardiomyocyte-associated methylation markers?
2. The authors used 0.5 ml of plasma. This amount appears to be rather small. Why did the authors not use a larger volume?
3. What are the age and sex distributions of the 83 healthy adult controls in Fig. 2A?
4. For the 31 patients with the clinical diagnosis of ST elevation myocardial infarction, the authors stated that they presented to the hospital within 12 hours of onset of acute chest pain. As the phrase 'within 12 hours' seemed to be imprecise time duration, could the authors be more specific and to provide more information regarding the number of subjects within more discrete time ranges?
5. Fig. 2E demonstrates a rather strong correlation between cardiac cfDNA and troponin levels, when the FAM101A concentration exceeds 100 copies/ml. However, in the Introduction section, the authors put forward the theoretical advantage of a DNA-based marker over a protein marker in that one might obtain a more quantitative readout. My question would therefore be: given such a high correlation between the cfDNA marker and the troponin marker, is the conjecture that a DNA marker would be more quantitative substantiated at all? Indeed, from the data presented for myocardial infarction, it would be difficult to argue for a convincing advantage of the DNA marker over a conventional protein marker. Actually, given the relative rapidity in which a protein marker can be assayed, not many diagnosticians would be easily swayed by the DNA-based approach.
6. For the sepsis work, the authors argue that as cfDNA is cleared by the liver, it may be superior to troponin. However, it is important to note that sepsis could also be associated with liver dysfunction. Hence, is it possible that the cardiomyocyte cfDNA marker levels might also be biased in patients with sepsis complicated by liver dysfunction? Had the authors measured the liver function tests for the cohort of patients with sepsis? The authors also seem to imply that cfDNA is not cleared by the kidneys. However, in the literature, there is clear evidence that fetal and tumor DNA can be found in the urine of pregnant women and non-urolgical cancer patients, respectively. The authors should thus discuss more about their proposition that cfDNA markers are less affected by the deterioration in renal function in septic patients.

7. In the various spike-in experiments performed by the authors, the results have been expressed in percentages. However, for the actual data generated using cfDNA, the authors have presented their data in copies/ml. I think that it would be more relevant if the spike-in experiments are also expressed as copies/ml. Indeed, it would be best if the spike-in cardiomyocyte DNA was added to plasma DNA previously shown to not to have detectable cardiomyocyte DNA.

Reviewer #3:

Remarks to the Author:

The authors show that circulating cell-free DNA (cfDNA) derived from dying myocardial cells is absent in the plasma of healthy individuals, but present in patients with acute myocardial infarction (which correlates with levels of troponin and CPK), and in patients with sepsis (where it is correlated with short-term mortality). In view of the limitations with the use of troponin, this information is welcome.

The data are interesting, and well discussed. The paper is well written. I have only a few minor comments

1. Throughout the paper: 'severe sepsis' is outdated; sepsis is always severe; please delete 'severe'.
2. The area under the curve (AUC) does not need 4 decimals.
3. 90-day mortality is not considered as 'short term' in the field of sepsis; please delete 'short term'.
4. Since all patients with myocardial infarction had STEMI, it would be preferable to use this abbreviation throughout the manuscript (instead of MI).

Reviewer #4:

Remarks to the Author:

[Redacted]

Detailed response to reviewer comments:

We thank the reviewers for their detailed and constructive comments. We have modified the manuscript as requested, including both experimental and textual changes, in particular rephrasing statements about the troponin assay.

Reviewer #1

Zemmour et al report their investigation of using cardiomyocyte-associated methylation signatures to develop plasma DNA markers for detecting human cardiomyocyte cell death. This work follows on publications by this team and other groups that circulating methylation markers may serve as laboratory markers for different clinical conditions.

I have the following specific comments.

1. Treatment with bisulfite has generally been known to be destructive to DNA. How significant is such degradation on the robustness of detecting circulating cardiomyocyte-associated methylation markers?

Indeed, bisulfite treatment leads to significant degradation of DNA. We quantified this loss using ddPCR to measure the amount of DNA in cfDNA preps before and after bisulfite treatment. We have designed primers to amplify the same region, adapted to the DNA sequence before and after conversion. In our hands, about 75% of cfDNA is lost upon bisulfite treatment. We tested n=9 plasma samples from 4 healthy donors; volume of plasma was 1-4 ml, and cfDNA was prepared using QiaAmp columns or Qiasymphony. The percentage of cfDNA lost ranged from 60 to 90. See results below; we do not think this belongs in this paper but are happy to do so if deemed necessary. We mention the findings on the extent of DNA degradation in the revised methods section (p.10).

This phenomenon does of course reduce assay sensitivity since most of the bisulfite-treated cfDNA is not available to PCR amplification (underscoring the importance of large blood volume for cfDNA methylation assays). However, the features of our assay minimize the effects of DNA degradation on assay performance. The assay measures the proportion of unmethylated DNA (of cardiac origin) among the molecules that survived bisulfite conversion. Since cardiac and non-cardiac cfDNA are equally affected by

bisulfite, this proportion is not sensitive to the extent of degradation. Note also that bisulfite is expected to convert all cytosines not adjacent to guanine. Taking advantage of this fact, our primers are designed to amplify only bisulfite-converted DNA. In addition, inefficient conversion is easily spotted in the sequencing data. The cardiac cfDNA proportion is then translated to absolute cardiac DNA copies / ml by multiplying with the total concentration of cfDNA; the latter is measured from the cfDNA of each sample prior to bisulfite conversion, again making it independent of bisulfite conversion efficiency or degradation. These considerations are explained briefly in the revised manuscript (p.10-11).

2. The authors used 0.5 ml of plasma. This amount appears to be rather small. Why did the authors not use a larger volume?

Thank you. When obtaining fresh samples, we draw 20 ml of blood, from which 8 ml of plasma were extracted. When using archived samples (some of the sepsis samples) we used whatever volume was available, but no less than 400 microliters. The specific information on plasma volumes used is:

MI sample volume: 1000 – 4000, average 3721 μ l

Sepsis sample volume: 400 – 3500, average 1804 μ l

Control sample volume: 800 – 4000, average 3043 μ l

We have clarified this issue in the revised text (methods section, p.9).

3. What are the age and sex distributions of the 83 healthy adult controls in Fig. 2A?

Age distribution of controls was 18 – 66 years, average 35.3. Sex distribution of controls was 44 females, 37 males, 2 information not available. This information is detailed in the revised methods section, p.9.

4. For the 31 patients with the clinical diagnosis of ST elevation myocardial infarction, the authors stated that they presented to the hospital within 12 hours of onset of acute chest pain. As the phrase 'within 12 hours' seemed to be imprecise time duration, could the authors be more specific and to provide more information regarding the number of subjects within more discrete time ranges?

Thank you. The key parameter here needed to assess assay performance is the time from chest pain to blood sampling that took place PRIOR to PCI. The time from chest pain to hospital admission matters less in this context. We have clarified the issue in the revised manuscript, and provide detailed information about the timing from chest pain to sampling pre-PCI. 19 patients were sampled before PCI, as follows: within 1hr from chest pain: n=3; 1-4 hrs: n=17; 4-8 hrs: n=5; 8-13 hrs: n=6. This information is presented in the revised methods section.

5. Fig. 2E demonstrates a rather strong correlation between cardiac cfDNA and troponin levels, when the FAM101A concentration exceeds 100 copies/ml. However, in the Introduction section, the authors put forward the theoretical advantage of a DNA-based marker over a protein marker in that one might obtain a more quantitative readout. My question would therefore be: given such a high correlation between the cfDNA marker and the troponin marker, is the conjecture that a DNA marker would be more quantitative substantiated at all? Indeed, from the data presented for myocardial infarction, it would be difficult to argue for a convincing advantage of the DNA marker over a conventional protein marker. Actually, given the relative rapidity in which a protein marker can be assayed, not many diagnosticians would be easily swayed by the DNA-based approach.

Indeed, the correlation between troponin and cardiac cfDNA in STEMI samples is strong, and supports the validity of the cfDNA assay. The argument in the introduction was referring to the theoretical possibility of deducing the absolute number of dying cardiomyocytes from cfDNA (given the known DNA content per cell), which is difficult to do when using a protein biomarker.

Once we have established the validity of the assay in the "positive control" setting of STEMI, we assessed cardiac cfDNA in situations where the significance of troponin is less clear (e.g. sepsis). In such cases, cardiac cfDNA may provide complementary information (e.g. to prove the presence of cardiac cell death). In addition, discrepancies in troponin and cfDNA values may inform on the time from injury to sampling, due to the different clearance rates of the two markers. This is likely the explanation to the lack of correlation between troponin and cardiac cfDNA in the sepsis samples (Figure 4B).

We do not propose that cardiac cfDNA is superior to troponin, and are certainly aware of the power of troponin, including assay rapidity. Our purpose is to demonstrate the potential of the cfDNA biomarker, and to suggest that in some settings it might be useful to measure cardiac cfDNA in addition to troponin,

obviously after extensive further validation studies. The text in introduction and discussion was modified to clarify these points.

6. For the sepsis work, the authors argue that as cfDNA is cleared by the liver, it may be superior to troponin. However, it is important to note that sepsis could also be associated with liver dysfunction. Hence, is it possible that the cardiomyocyte cfDNA marker levels might also be biased in patients with sepsis complicated by liver dysfunction? Had the authors measured the liver function tests for the cohort of patients with sepsis? The authors also seem to imply that cfDNA is not cleared by the kidneys. However, in the literature, there is clear evidence that fetal and tumor DNA can be found in the urine of pregnant women and non-urolological cancer patients, respectively. The authors should thus discuss more about their proposition that cfDNA markers are less affected by the deterioration in renal function in septic patients.

This is an important point. We measured circulating liver enzymes (AST and ALT) in 41 sepsis samples, and correlated the levels to cardiac (and total) cfDNA levels. There was no correlation between liver damage and cardiac or total cfDNA, suggesting that liver dysfunction is not a major confounder of cardiac cfDNA data.

With regard to cfDNA clearance by the kidney: the major route of cfDNA clearance was shown to be the liver (removing ~80% of injected nucleosomes) with only a minor fraction (0.5%) removed by the kidney (Gauthier et al 1996, reference 22 in the manuscript). In addition, patients with kidney dysfunction do not show a significant elevation in cfDNA levels:

1. Dwivedi, D.J., Toltl, L.J., Swystun, L.L., Pogue, J., Liaw, K.L., Weitz, J.I., Cook, D.J., Fox-Robichaud, A.E. and Liaw, P.C., 2012. Prognostic utility and characterization of cell-free DNA in patients with severe sepsis. *Critical Care*, 16(4), p.R151.

2. McGuire, A.L., Urosevic, N., Chan, D.T., Dogra, G., Inglis, T.J. and Chakera, A., 2014. The impact of chronic kidney disease and short-term treatment with rosiglitazone on plasma cell-free DNA levels. *PPAR research*, 2014:643189.

The effects of kidney and liver dysfunction on cfDNA levels are discussed in more detail in the revised manuscript (p.6), and we have added a new supplementary Figure (S5) showing that liver or kidney damage does not necessarily lead to elevated cardiac cfDNA.

7. In the various spike-in experiments performed by the authors, the results have been expressed in percentages. However, for the actual data generated using cfDNA, the authors have presented their data in copies/ml. I think that it would be more relevant if the spike-in experiments are also expressed as copies/ml. Indeed, it would be best if the spike-in cardiomyocyte DNA was added to plasma DNA previously shown to not to have detectable cardiomyocyte DNA.

The purpose of the spike-in experiments was to demonstrate the identification of cardiac DNA when present as a **small fraction** in non-cardiac DNA, hence the choice of percentage and not copy number, which we have elected to retain in Figure 1.

One may also test **the smallest number of cardiac DNA copies that can be detected**, regardless of dilution in other DNA. We performed such experiments, and found that we can detect down to 3 cardiac genome equivalents, although when the number of cardiac genomes present in the tube is below 10, results become less reproducible. We believe that this is because of bisulfite-related degradation, often leaving zero copies of the target in the tube. The results of these experiments are shown as a new panel D in Supplemental Figure S1 and described in the revised text.

In addition, we modified Figure 5C (ddPCR spike in experiment) to show how both % input and copy number input affect the outcome. ddPCR captures about a third of input molecules, likely due to loss of many molecules to bisulfite.

Regarding to proposal to mix genomic cardiac DNA with plasma: these are very different entities. Genomic cardiac DNA comes in big pieces while plasma contains short fragments of cfDNA.

Reviewer #3

The authors show that circulating cell-free DNA (cfDNA) derived from dying myocardial cells is absent in the plasma of healthy individuals, but present in patients with acute myocardial infarction (which correlates with levels of troponin and CPK), and in patients with sepsis (where it is correlated with short-term mortality). In view of the limitations with the use of troponin, this information is welcome. The data are interesting, and well discussed. The paper is well written.

I have only a few minor comments

1. Throughout the paper: 'severe sepsis' is outdated; sepsis is always severe; please delete 'severe'.

Done.

2. The area under the curve (AUC) does not need 4 decimals.

Changed to have only 2 decimals.

3. 90-day mortality is not considered as 'short term' in the field of sepsis; please delete 'short term'.

Done.

4. Since all patients with myocardial infarction had STEMI, it would be preferable to use this abbreviation throughout the manuscript (instead of MI).

Done.

Reviewer #4

[Redacted]

Reviewer #1:

Remarks to the Author:

The authors have satisfactorily addressed my previous comments.

Reviewer #4:

Remarks to the Author:

[Redacted]

Response to reviewer comments:

Reviewer #4

[Redacted]